# Representation Learning in Continuous-Time Score-Based Generative Models

Korbinian Abstreiter [1]  Stefan Bauer [2]  Arash Mehrjou [2] [1]

## Abstract

Score-based methods represented as stochastic differential equations on a continuous time domain have recently proven successful as a non-adversarial generative model. Training such models relies on denoising score matching, which can be seen as multi-scale denoising autoencoders. Here, we augment the denoising score-matching framework to enable representation learning without any supervised signal. GANs and VAEs learn representations by directly transforming latent codes to data samples. In contrast, score-based representation learning relies on a new formulation of the denoising score-matching objective and thus encodes information needed for denoising. We show how this difference allows for manual control of the level of detail encoded in the representation.

## 1. Score-based generative modeling

Score-based methods have recently proven successful for generating images (Song & Ermon, 2020; Song et al., 2020), graphs (Niu et al., 2020), shapes (Cai et al., 2020), and audio (Chen et al., 2020b; Kong et al., 2021). Two promising approaches apply step-wise perturbations to samples of the data distribution until the perturbed distribution matches a known prior (Song & Ermon, 2019; Ho et al., 2020). A model is trained to estimate the reverse process, which transforms samples of the prior to samples of the data distribution. These diffusion models have been further refined (Nichol & Dhariwal, 2021; Jolicoeur-Martineau et al., 2020; Luhman & Luhman, 2021) and even achieved better image sample quality than GANs (Dhariwal & Nichol, 2021). Further, Song et al. showed that these frameworks are discrete versions of continuous-time perturbations by stochastic differential equations and propose a score-based generative modeling

[1]ETH Zürich [2]Max Planck Institute for Intelligent Systems. Correspondence to: Korbinian Abstreiter <kabstreiter@ethz.ch>, Stefan Bauer <stefan.bauer@tuebingen.mpg.de>, Arash Mehrjou <amehrjou@ethz.ch>.

Third workshop on *Invertible Neural Networks, Normalizing Flows, and Explicit Likelihood Models* (ICML 2021). Copyright 2021 by the author(s).

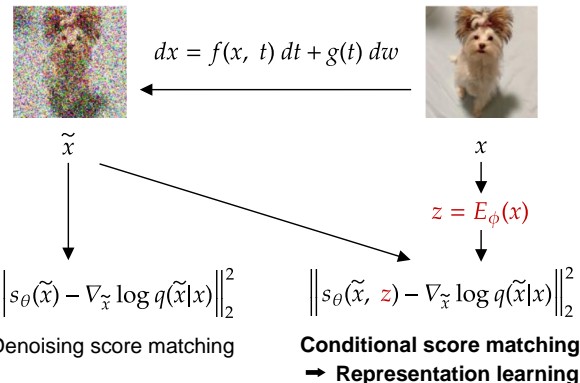

Figure 1. **Conditional score matching with a parametrized latent code is representation learning.** Denoising score matching estimates the score at each $\tilde{x}$; we add a latent representation $z$ of the clean data $x$ as additional input to the score estimator.

framework on continuous time.

Learning desirable representations has been an inseparable component of generative models such as GANs and VAEs (Radford et al., 2016; Chen et al., 2016; Higgins et al., 2017; Burgess et al., 2018; van den Oord et al., 2017; Donahue & Simonyan, 2019; Chen et al., 2020a). Considering score-based methods as promising and theoretically grounded generative models, here we propose a method to augment their underlying SDE for learning a latent data-generating code. The key idea of our approach is illustrated in Figure 1. We begin by briefly revisiting the foundations of score-based generative diffusion models in section 1.1. In section 2 we present our method and follow up with experimental results in section 3.

### 1.1. Forward and reverse diffusion process

The forward diffusion process of the data is modeled as a Stochastic Differential Equation (SDE) on a continuous time domain $t \in [0, T]$. Let $x_0 \in \mathcal{R}^d$ denote a sample of the data distribution $x_0 \sim p_0$, where $d$ is the data dimension. The trajectory $(x_t)_{t \in [0,T]}$ of data samples is a function of time determined by the stochastic diffusion process. The SDE is chosen such that the distribution $p_{0T}(x_T|x_0)$ for any sample $x_0 \sim p_0$ can be approximated by a known prior distribution. Notice that the subscript $0T$ of $p_{0T}$ refers to the conditional distribution of the diffused data at time $T$ given

the data at time 0. For simplicity we limit the remainder of this paper to the so-called Variance Exploding SDE (Song et al., 2021), which is defined as

$$dx = f(x,t)\, dt + g(t)\, dw := \sqrt{\frac{d[\sigma^2(t)]}{dt}}, \qquad (1)$$

where w is the standard Wiener process. The perturbation kernel of this diffusion process has a closed-form solution being $p_{0t}(x_t|x_0) = \mathcal{N}(x_t; x_0, [\sigma^2(t) - \sigma^2(0)]I)$. It was shown by Anderson that the reverse diffusion process is the solution to the following SDE:

$$dx = [f(x,t) - g^2(t)\nabla_x \log p_t(x)]\, dt + g(t)\, d\overline{w}, \quad (2)$$

where $\overline{w}$ is the standard Wiener process on reverse time flow. Thus, given the score function $\nabla_x \log p_t(x)$ for all $t \in [0, T]$, we can generate samples from the data distribution $p_0(x)$.

## 1.2. Denoising score matching objective

In order to learn the score function, one would like to minimize the distance between the model and the true score function. This method is called Explicit Score Matching (Vincent, 2011) and has the following objective function:

$$J_t^{ESM}(\theta) = \mathbf{E}_{x_t}\left[\|s_\theta(x_t, t) - \nabla_{x_t} \log p_t(x_t)\|_2^2\right]. \quad (3)$$

Since the ground-truth score function $\nabla_{x_t} \log p_t(x_t)$ is generally not known, one can apply denoising score matching (Vincent, 2011), which is defined as the following:

$$J_t(\theta) = \mathbf{E}_{x_0}\{\mathbf{E}_{x_t|x_0}[$$
$$\|s_\theta(x_t, t) - \nabla_{x_t} \log p_{0t}(x_t|x_0)\|_2^2]\}. \quad (4)$$

The issue of single scale noise motivated Song & Ermon to expand the objective to a sum of denoising score matching terms on multiple noise scales. They further augment the objective with a positive weighting function $\lambda(\sigma) > 0$ to empirically balance the loss magnitudes for all noise levels. For the continuous time domain, Song et al. uniformly sample $t \in [0, T]$ and use a time-dependent positive weighting function $\lambda(t)$, leading to the following objective:

$$J(\theta) = \mathbf{E}_t\left[\lambda(t) J_t(\theta)\right]. \quad (5)$$

We now show that this objective cannot be made arbitrarily small. It is known that (4) is equal to explicit score matching up to a constant which is independent of $\theta$ (Vincent, 2011). Thus, the objective is minimized when the model equals the ground-truth score function $s_\theta(x_t, t) = \nabla_x \log p_t(x)$ and the additional constant is equal to the loss when this equality holds. This leads to the following new formulation of the denoising score matching objective:

$$J_t(\theta) = \mathbf{E}_{x_0}\{\mathbf{E}_{x_t|x_0}[$$
$$\|\nabla_{x_t} \log p_{0t}(x_t|x_0) - \nabla_{x_t} \log p_t(x_t)\|_2^2 \quad (6)$$
$$+ \|s_\theta(x_t, t) - \nabla_{x_t} \log p_t(x_t)\|_2^2]\}.$$

This observation has not been emphasized previously, probably because it has no direct effect on the learning of the score function. However, the additional constant has major implications for finding other hyperparameters. Examples for such hyperparameters are the values of: the function $\lambda(t)$ and the choice of the forward SDE. While these hyperparameters could be optimized in explicit score matching using gradient-based learning, this ability is severely limited by the additional non-vanishing constant in (6). In particular, optimizing such hyperparameters based on the denoising score matching objective leads to solutions that do not necessarily minimize the distance from the model to the ground-truth score function. Instead, they are heavily biased towards solutions with a smaller value of the additional constant. For example, trying to minimize the worst-case $\lambda$-divergence as defined in (Durkan & Song, 2021) with an adversarially trained $\lambda$ is not directly possible, since $\lambda$ will focus on regions where the constant is high and mostly ignores the model fit to the ground-truth score.

The non-vanishing constant in the denoising score matching objective, which presents a burden in multiple ways such as hyperparameter search and model evaluation, however also provides an opportunity for latent representation learning, which will be described in the following sections.

# 2. Representation learning through score-matching

## 2.1. Conditional score matching

Class-conditional generation can be achieved in this framework by training an additional time-dependent classifier $p_t(y|x_t)$ (Song et al., 2021). In particular, the conditional score for a fixed $y$ can be expressed as the sum of the unconditional score and the score of the classifier, that is,

$$\nabla_{x_t} \log p_t(x_t|y) = \nabla_{x_t} \log p_t(x_t) + \nabla_{x_t} \log p_t(y|x_t).$$

We propose conditional score matching as an alternative way to allow for controllable generation. Given supervised labels $y(x)$, the new training objective for each time $t$ becomes

$$J_t(\theta) = \mathbf{E}_{x_0}\{\mathbf{E}_{x_t|x_0}[$$
$$\|s_\theta(x_t, t, y(x_0)) - \nabla_{x_t} \log p_{0t}(x_t|x_0)\|_2^2]\}. \quad (7)$$

The conditional objective is minimized if and only if the model equals the conditional score function $\nabla_{x_t} \log p_t(x_t|y(x_0) = \hat{y})$ for all labels $\hat{y}$. Note that conditional score matching is directly done during training and does not require to train an additional classifier over the whole time domain.

## 2.2. Learning the latent representation

Since supervised data is limited and rarely available, we propose to learn the labeling function $y(x_0)$ at the same time as

optimizing the conditional score matching objective (7). In particular, we represent the labeling function as a trainable encoder $E_\phi : \mathcal{R}^d \to \mathcal{R}^{d_z}$, where $E_\phi(x_0)$ maps the data sample $x_0$ to its corresponding code in the $d_z$-dimensional latent space. The code is then used as additional input to the model. Formally, the proposed learning objective for latent representation learning is the following:

$$J(\theta, \phi) = \mathbf{E}_{t,x_0,x_t}[\lambda(t)$$
$$\|s_\theta(x_t, t, E_\phi(x_0)) - \nabla_{x_t} \log p_{0t}(x_t|x_0)\|_2^2]. \quad (8)$$

Intuitively, $E_\phi(x_0)$ selects the vector field used to denoise $x_0$ starting from $x_t$. We show in the following that (8) is a valid representation learning objective. The score of the perturbation kernel $\nabla_{x_t} \log p_{0t}(x_t|x_0)$ is a function of only $t$, $x_t$ and $x_0$. Thus the objective can be reduced to zero if all information about $x_0$ is contained in the latent representation $E_\phi(x_0)$. When $E_\phi(x_0)$ has no mutual information with $x_0$, the objective can only be reduced up to the constant in (6). Hence, our proposed formulation takes advantage of the non-zero lower-bound of (6) which can only vanish when data information is distilled in a code provided as input to the model.

## 2.3. Controlling the representation

In contrast to other methods used for unsupervised representation learning (Radford et al., 2016; Chen et al., 2016; Higgins et al., 2017), the proposed objective here enjoys the continuous nature of the SDE. The encoder is trained to represent information needed to denoise $x_0$ for different levels of noise $\sigma(t)$. We hypothesize that by adjusting the weighting function $\lambda(t)$, we can manually control the granularity of the features encoded in the representation. For high noise levels, the mutual information of $x_t$ and $x_0$ is insignificant, thus denoising requires all information about $x_0$ to be contained in the code. In contrast, for small values of $t$, $x_t$ still contains coarse-grained features of $x_0$ and denoising can be performed even when the representation encodes only fine-grained features. We provide empirical evidence to support this hypothesis in Section 3.

# 3. Experimental results

For all experiments, we use the same function $\sigma(t), t \in [0, 1]$ as in (Song et al., 2021), which is $\sigma(t) = \sigma_{\min} \left( \frac{\sigma_{\max}}{\sigma_{\min}} \right)^t$, where $\sigma_{\min} = 0.01$ and $\sigma_{\max} = 50$. Further, we use $\lambda(t) = \sigma^2(t)$, which has been shown to yield the KL-Divergence objective (Durkan & Song, 2021). For visualization purposes, we use a 2-dimensional latent space if not stated otherwise. Our goal is not to produce state-of-the-art image quality, rather showcase the representation learning method. Because of that and also limited computational resources, we did not carry out an extensive hyper-parameter


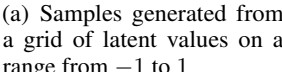
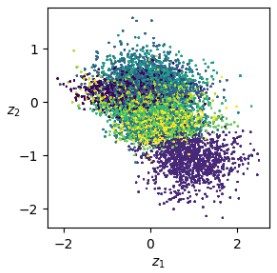

(a) Samples generated from a grid of latent values on a range from $-1$ to $1$

(b) Latent representation of test samples, colored according to the digit class

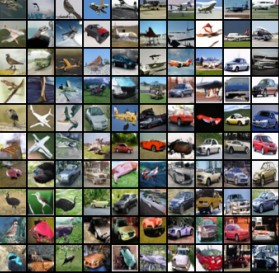
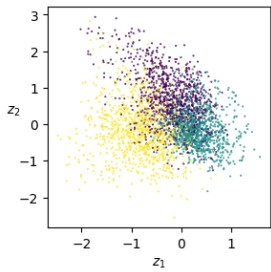

(c) Samples generated from a grid of latent values on a range from $-1$ to $1$

(d) Latent representation of test samples, colored according to the class label

Figure 2. Samples and latent distribution of a model trained on MNIST (a-b) and the first three classes of CIFAR-10 (c-d) using L1-regularization and uniform sampling of $t$

sweep. Hence, the model architecture in all experiments is similar to but significantly smaller than the one proposed in (Song et al., 2021). Details for architecture and hyperparameters are described in the appendix (A.1). Figure 10 in the appendix further illustrates how the representation encodes information for denoising.

## 3.1. Uniform sampling of $t$

We first train a model using L1-regularization on the latent code for the MNIST dataset (LeCun & Cortes, 2010) and CIFAR-10 (Krizhevsky et al.). Due to computational limitations, we limit CIFAR-10 to a subset of only three classes, which we randomly chose to be the first three classes. Figure 2 shows samples from a grid over the latent space and a point cloud visualization of the latent values $z = E_\phi(x_0)$. For MNIST, we can see that the value of $z_1$ controls the stroke width, while $z_2$ weakly indicates the class. In contrast, the latent code of CIFAR-10 samples mostly encodes information about the class label. We can see from the samples that part of the reason might be an encoding of the background, which is highly correlated with the class labels. We conducted the same experiment with a probabilistic encoder, where the latent representation is regularized using

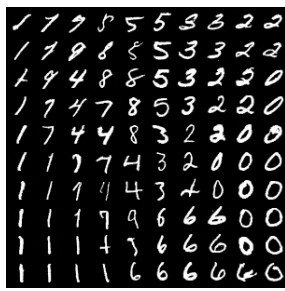 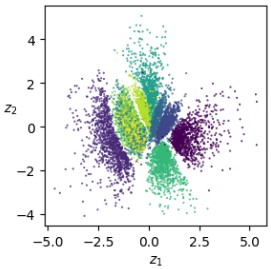

(a) Samples generated from a grid of latent values on a range from $-2$ to $2$

(b) Latent representation of test samples, colored according to the digit class

*Figure 3.* Samples and latent distribution of a model trained on MNIST using KL-divergence and uniform sampling of $\sigma$

**KL-Divergence.** The resulting representation is similar and can be seen in the appendix (5, 6). We also trained models on all classes of CIFAR-10, however not until convergence due to computational constraints (cf. 8, 9). Early results indicate encoding of overall image brightness. We further want to point out that the generative process using the reverse SDE involves randomness and thus can generate different samples for a single latent representation. The diversity of samples generated for the same representation steadily decreases with the dimensionality of the latent space, which is empirically shown in Figure 11 of the appendix.

### 3.2. Controlling the representation

Next, we analyze the behavior of the representation when adjusting the weighting function $\lambda(t)$, which can be done by changing the sampling distribution of $t$.

#### 3.2.1. HIGH NOISE LEVELS

First, we focus the training on higher noise levels. To this end, we sample $t$ such that $\sigma(t)$ is uniformly sampled from the interval $[\sigma_{\min}, \sigma_{\max}] = [0.01, 50]$. Note that after learning the representation we additionally train the model with uniform sampling of $t$ and a frozen encoder to achieve good sample quality. Figure 3 shows the resulting representation for MNIST using a probabilistic encoder (cf. Figure 7 for L1 regularization results). As expected, the latent representation encodes information about classes rather than fine-grained features such as stroke width. This validates our hypothesis of section 2.3 that we can control the granularity of features encoded in the latent space.

#### 3.2.2. TRAINING ON SINGLE TIMESCALES

To understand the effect of training on different timescales more clearly, we limit the support of the weighting function $\lambda(t)$ to a single value of $t$. We analyze the resulting quality of the latent representation for different values of $t$ using the silhouette score with euclidean distance based on the

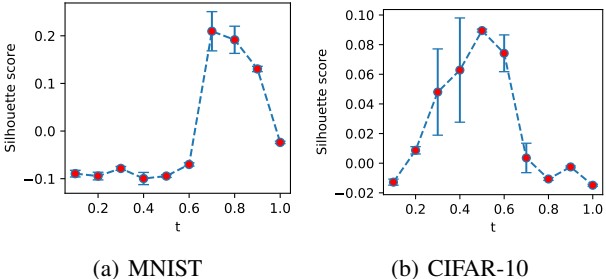

(a) MNIST

(b) CIFAR-10

*Figure 4.* Mean and standard deviation of silhouette scores when training a model on MNIST (left) and the first three classes of CIFAR-10 (right) using a single $t$ over three runs.

dataset classes (Rousseeuw, 1987). It compares the average distance between a point to all other points in its cluster with the average distance to points in the nearest different cluster. Thus we measure how well the latent representation encodes classes, ignoring any other features.

Figure 4 shows the silhouette scores of latent codes of MNIST and CIFAR-10 samples for different values of $t$. In alignment with our hypothesis of section 2.3, training on a small $t$ and thus low noise levels leads to almost no encoded class information in the latent representation, while the opposite is the case for a range of $t$ which differs between the two datasets. The decline in encoded class information for high values of $t$ can be explained by the vanishing difference between distributions of perturbed samples when $t$ gets large. This shows that the distinction among the code classes represented by the silhouette score is controlled by $\lambda(t)$.

Overall, the difference in the latent codes for varying $\lambda(t)$ shows that we can control the granularity encoded in the representation. While this ability does not exist in previously proposed models for representation learning, it provides a significant advantage when there exist some prior information about the level of detail that we intend to encode in the target representation.

## 4. Conclusion

We presented a new objective for representation learning based on conditional denoising score matching. In doing so, we turned the original non-vanishing objective function into one that can be reduced to zero if all information is distilled in the code. We showed that the proposed method learns interpretable features in the latent space. In contrast to previous approaches, denoising score matching as a foundation comes with the ability to manually control the granularity of features encoded in the representation. We demonstrated that the encoder can learn to separate classes when focusing on high noise levels and encodes fine-grained features such as stroke-width when mainly trained on low level noise.

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

# A. Appendix

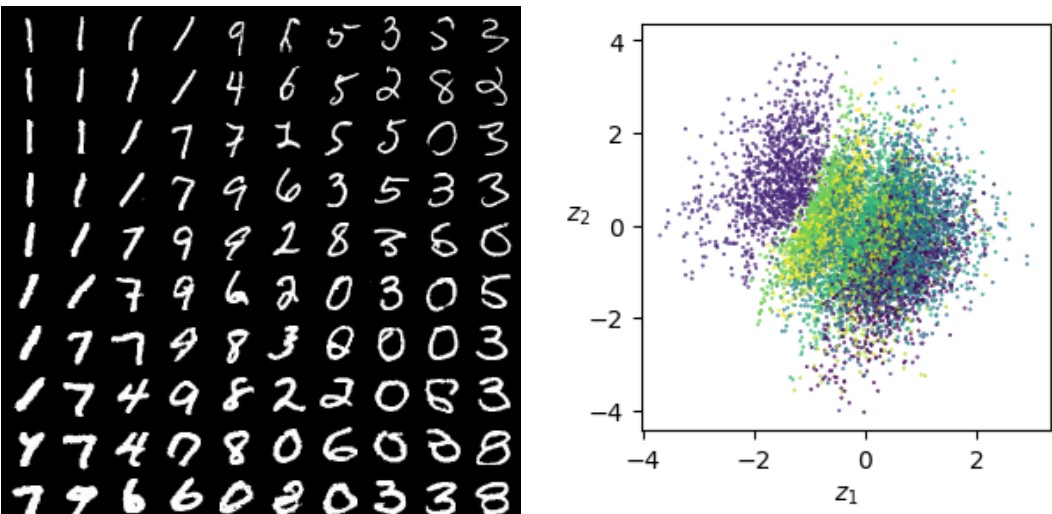

(a) Samples generated from a grid of latent values on a range from $-2$ to $2$

(b) Latent representation of test samples, colored according to the digit class

*Figure 5.* Samples and latent distribution of a model trained on MNIST using KL-divergence and uniform sampling of $t$

## A.1. Architecture and Hyperparameters

The model architecture we use for all experiments is based on "DDPM++ cont. (deep)" used for CIFAR-10 in (Song et al., 2021). It is composed of a downsampling and an upsampling block with residual blocks at multiple resolutions. We did not change any of the hyperparameters of the optimizer. Depending on the dataset, we adjusted the number of resolutions, number of channels per resolution, and the number of residual blocks per resolution in order to reduce training time.

For representation learning, we use an encoder with the same architecture as the downsampling block of the model, followed by another three dense layers mapping to a low dimensional latent space. Another four dense layers map the latent code back to a higher-dimensional representation. It is then given as input to the model in the same way as the time embedding. That is, each channel is provided with a conditional bias determined by the representation and time embedding at multiple stages of the downsampling and upsampling block.

**Regularization of the latent space**   For both datasets, we use a regularization weight of $10^{-5}$ when applying L1-regularization, and a weight of $10^{-7}$ when using a probabilistic encoder regularized with KL-Divergence.

**MNIST hyperparameters**   Due to the simplicity of MNIST, we only use two resolutions of size $28 \times 28 \times 32$ and $14 \times 14 \times 64$, respectively. The number of residual blocks at each resolution is set to two. In each experiment, the model is trained for $80k$ iterations. For uniform sampling of $\sigma$ we trained the models for an additional $80k$ iterations with a frozen encoder and uniform sampling of $t$.

**CIFAR10 hyperparameters**   For the silhouette score analysis, we use three resolutions of size $32 \times 32 \times 32$, $16 \times 16 \times 32$, and $8 \times 8 \times 32$, again with only two residual blocks at each resolution. Each model is trained for $90k$ iterations.

**CIFAR10 (deep) hyperparameters**   While representation learning works for small models already, sample quality on CIFAR-10 is poor for models of the size described above. Thus for models used to generate samples, we use eight residual blocks per resolution and the following resolutions: $32 \times 32 \times 32$, $16 \times 16 \times 64$, $8 \times 8 \times 64$, and $4 \times 4 \times 64$. Each model is trained for $300k$ iterations.

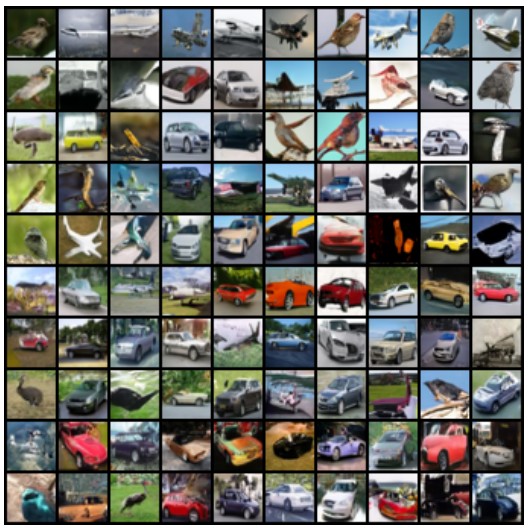 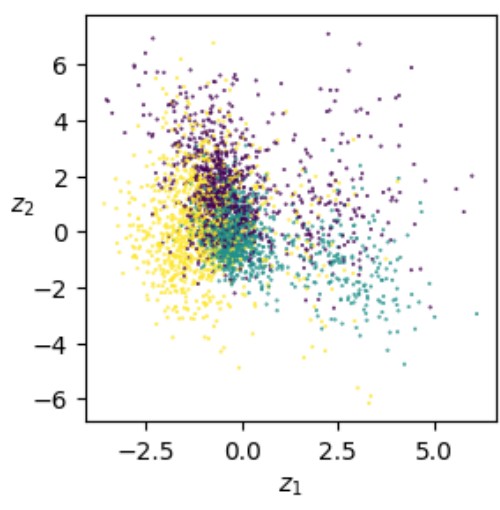

(a) Samples generated from a grid of latent values on a range from $-1$ to $1$

(b) Latent representation of test samples, colored according to the class label

*Figure 6.* Samples and latent distribution of a model trained on the first three classes of CIFAR-10 using KL-divergence and uniform sampling of $t$

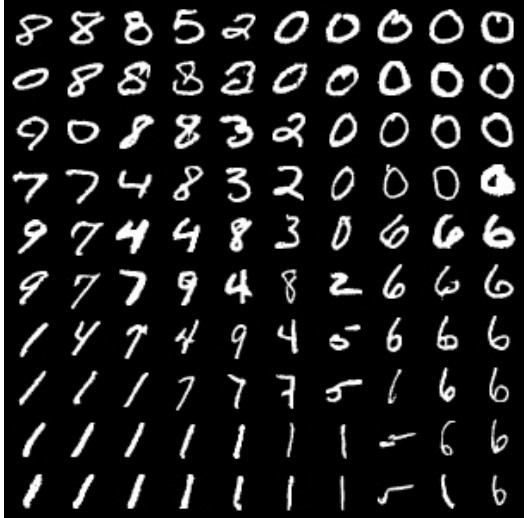 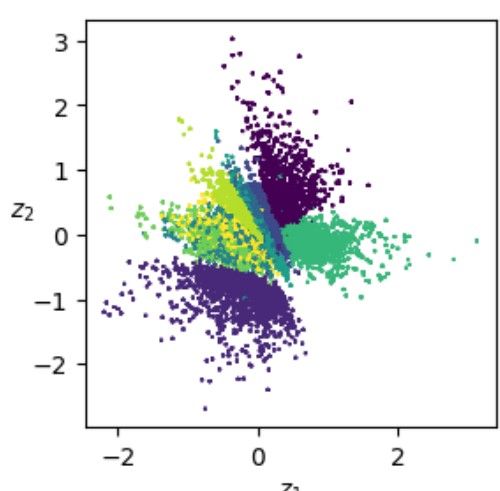

(a) Samples generated from a grid of latent values on a range from $-1$ to $1$

(b) Latent representation of test samples, colored according to the digit class

*Figure 7.* Samples and latent distribution of a model trained on MNIST using L1-regularization and uniform sampling of $\sigma$

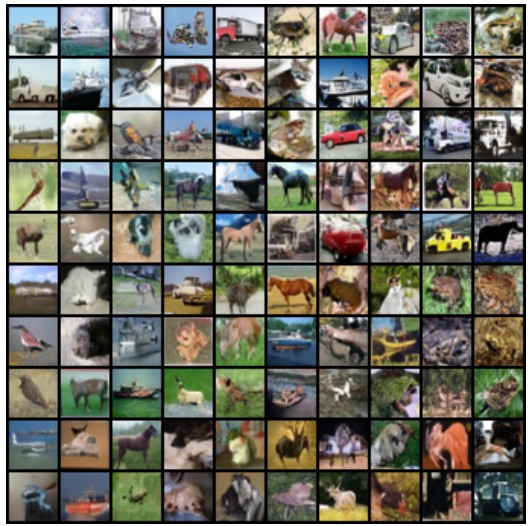 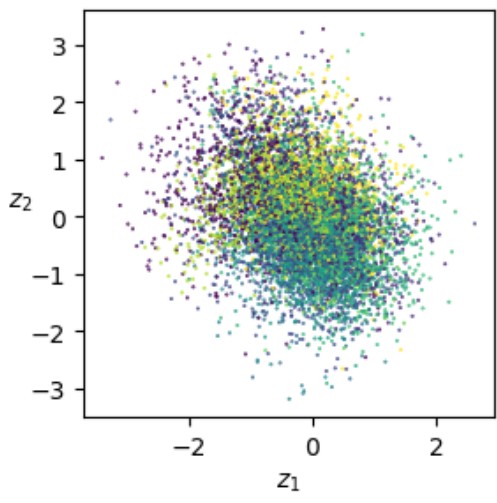

(a) Samples generated from a grid of latent values on a range from $-1$ to $1$

(b) Latent representation of test samples, colored according to the class label

*Figure 8.* Samples and latent distribution of a model trained on all classes of CIFAR-10 using KL-divergence and uniform sampling of $t$

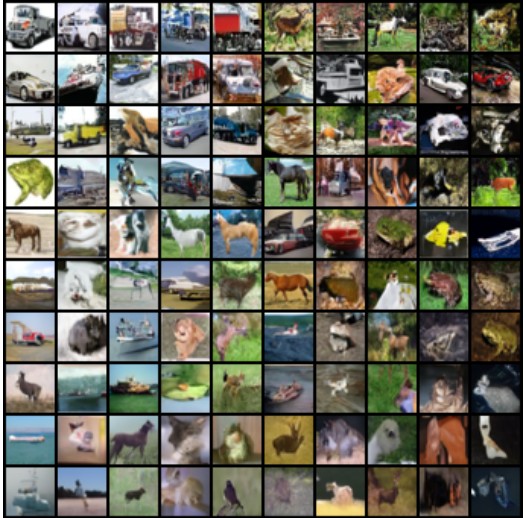 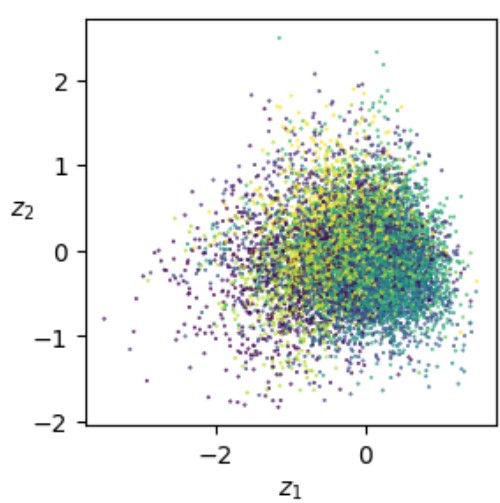

(a) Samples generated from a grid of latent values on a range from $-1$ to $1$

(b) Latent representation of test samples, colored according to the class label

*Figure 9.* Samples and latent distribution of a model trained on all classes of CIFAR-10 using L1-regularization and uniform sampling of $t$

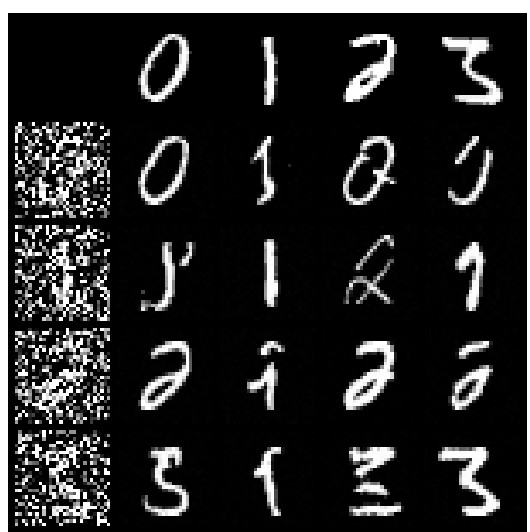

*Figure 10.* Samples generated starting from $x_t$ (left column) using the score function with the latent code of another $x_0$ (top row) as input. It shows that samples are denoised correctly only when conditioning on the latent code of the corresponding original image $x_0$.

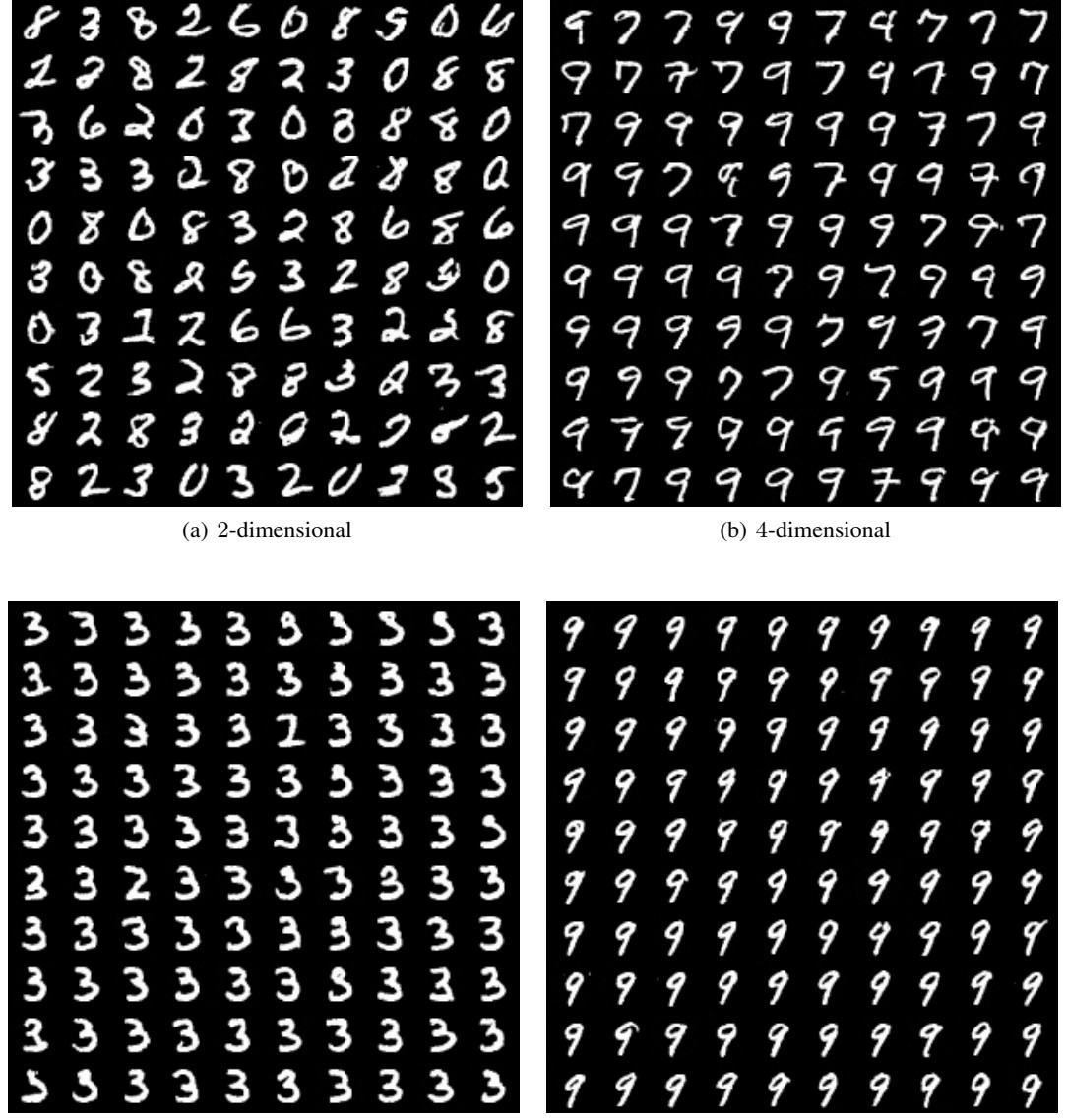

(a) 2-dimensional

(b) 4-dimensional

(c) 8-dimensional

(d) 16-dimensional

*Figure 11.* Samples generated using the same latent code for each generation, showing that the randomness of the code-conditional generation reduces in higher dimensional latent spaces.