# OpenReview forum: "Representation Learning in Continuous-Time Score-Based Generative Models"
_ICML.cc/2021/Workshop/INNF — INNF+ 2021 poster_

### Official Review · Reviewer_7puz · 2021-06-10

**Rating:** Accept
**Confidence:** 3

**Summary:**

This paper introduces a method for doing representation learning with continuous-time score based generative models. The approach is based on conditioning the score on a learned representation of the input. The paper motivates this approach by formulating the score matching objective as a sum of two terms. In particular, one of the two terms is irreducible in unconditional score matching but could be reduced in conditional score matching if the conditioning quantity encodes the observations. The representation should then allow reducing this term to zero. Applying different temporal weighting to the objective could encourage various level representations from semantic to fine-grained details. Results validate the approach.

**Justification For Rating:**

- The idea behind the paper seems at first simplistic but is in fact well-motivated by section 2. Making this section slightly stronger and more formal would improve the manuscript.
- Experimenting on the weighting function would be interesting and comparing the effect of using a continuous-time model to other generative models would be interesting as well. Indeed, it is not completely clear to me what is the advantage of using such representation learning instead of a VAE trained on different noise levels.
Although stong motivations for using the proposed method instead of others is not really demonstrated in the experiments I am sure there are and this should just be emphasized if this work is continued. Overall this short paper was very pleasant to read.

---

### Official Review · Reviewer_CEhZ · 2021-06-11

**Rating:** Accept
**Confidence:** 3

**Summary:**

Following work by Song et al on "Score Based Generative Models Through SDE", the authors suggest to learn a latent representation that is then used as a conditioning variable inside the score matching objective. This representation learning is unsupervised. The authors then use simple experiments to demonstrate some of the properties of their learnt representation.

**Justification For Rating:**

Paper is well written, and experiments demonstrate both the whole model setup, as well as studying the effects of some components independently.

There are multiple redefinitions of J_t(\theta) in equations 4, 6 and 7. I wonder if it would be worth using slightly different names to help differentiate them (and it's not clear they actually all need a name).

---

### Decision · Program_Chairs · 2021-06-14

Accept (poster)